# Structure and reactivity of germylene-bridged digold complexes

Liliang Wang [1], Guorong Zhen[1], Yinhuan Li[1], Mitsuo Kira[1✉], Liping Yan[1], Xiao-Yong Chang [2✉], Lu Huang[1] & Zhifang Li [1✉]

The bonding between gold and main-group metallic elements (M) featuring $Au^{\delta-}-M^{\delta+}$ polarity, has been studied recently. The gold in the bonds is expected to have the oxidation number of −1, and hence, nucleophilic. However, the knowledge of the reactivity of the gold-metal bonds remains limited. Here, we report digold-substituted germanes of the form of $R'_2Ge(AuPR_3)(AuGeR'_2)$ (**3a**; R = Me, **3b**; R = Et), featuring two Au-Ge(IV) and one Au-Ge(II) bonds. DFT calculations of **3a** revealed the existence of high-lying σ(Ge-Au) type HOMO and low-lying LUMO with germylene $p_\pi$ nature. A pendular motion of $AuPR_3$ group between Ge(IV) and Ge(II) of **3** occurs in the NMR time scale, suggesting that the Ge(II) center has an enhanced electrophilicity to be attacked by the nucleophilic gold (−I) atom. **3a** reacts with nucleophilic $Cl^-$ and electrophilic MeOTf reagents at Ge(II) and Ge(IV) centers, respectively.

[1] College of Material, Chemistry and Chemical Engineering, Key Laboratory of Organosilicon Chemistry and Material Technology, Ministry of Education, Hangzhou Normal University, 311121 Hangzhou, Zhejiang, People's Republic of China. [2] Department of Chemistry, Southern University of Science and Technology, 518055 Shenzhen, Guangdong, People's Republic of China. ✉email: mitsuo.kira.e2@tohoku.ac.jp; changxy@sustech.edu.cn; zhifanglee@hznu.edu.cn

Among transition metal elements, gold is unique with the large electronegativity ($\chi = 2.54$) and high electron affinity (2.30 eV) owing to the large relativistic effects[1]. Auride ion Au[−] has been known as a halogen-like anion with the electronic configuration of $5d^{10}6s^2$ since the discovery of caesium auride by Sommer in 1943[2].

The chemistry of gold compounds bonded to main-group elements has been developed extensively in recent years[3–7]. Goicoechea, Aldridge and coworkers have reported that novel gold-aluminum complex I (Fig. 1) has a nucleophilic auride character due to its $Al^{\delta+}$–$Au^{\delta-}$ bond polarity and reacts with $CO_2$ and a carbodiimide giving the corresponding insertion products[6]. Borylgold complexes II and III (Fig. 1) have been synthesized by Nozaki et al. and Kinjo et al., respectively, using the reactions of the corresponding boryllithiums with $Ph_3PAuCl$[4,5].

During our study of unique cyclic $(R_2SnAu)_3$ complex[8], we have discovered that digoldstannane IV is obtained by the reduction of the corresponding gold-substituted tin chloride V with $KC_8$ followed by the addition of excess $PEt_3$ (Fig. 1). Compounds with a gold–germanium bond may be interesting because the bond is less polarized than a gold-tin bond due to larger electronegativity of germanium ($\chi = 2.01$) than silicon ($\chi = 1.90$) and tin ($\chi = 1.96$)[1,9]. Although a variety of compounds with gold–germanium bonds have been studied since the first report of Glockling and Hooton in 1962[10], knowledge of the chemistry including the bonding characteristics and reactivities is still limited in gold-monosubstituted germanes[11–18] and germylene gold complexes[19–28]. While the preparation of isolable digoldgermane with Au(-I)–Ge bonds remains challenging, a number of single-atom bridged polygold complexes including those bridged with hydride[29–32], carbon[33–39], nitrogen[40–45], oxygen[46,47], and halogen[33,48–52] are known and widely utilized in the homogenous catalysis[53] and materials science[29,54].

In this work, we report the synthesis and properties of digoldgermanes 3 (3a: R = Me; 3b: R = Et) that feature two gold atoms coordinated by dialkylgermylene 1 and trialkylphosphine respectively (Fig. 2a). The structural characteristics of 3 were elucidated using NMR spectroscopy, X-ray crystallography and density-functional theory (DFT) calculations. The discussion is focused mainly on structural characteristics of the two different types of Au–Ge bonding of 3, fluctuation of the $AuPR_3$ group between $^1Ge$ and $^2Ge$ atoms of 3 in solution, and their distinctive reactions and catalysis.

## Results

### Synthesis and structural elucidation of digoldgermanes 3a (R = Me) and 3b (R = Et).

Digoldgermanes 3a and 3b are synthesized by applying the reaction route shown in Fig. 2a; see the SI for the experimental details. The reactions of an isolable dialkylgermylene 1 with $R_3PAuCl$ (R = Me and Et) in tetrahydrofuran (THF) at ambient temperature give 2a (R = Me) and

2b (R = Et), respectively, as white solids in almost quantitative yields; many similar insertion reactions of tetrylenes into Au–X bonds have been reported[13,27]. When 2a and 2b are treated with potassium graphite ($KC_8$) in THF at room temperature, the corresponding digoldgermanes 3a and 3b are obtained as dark-green solids in 49% and 54% yields, respectively. Compounds 3a and 3b are isolated as pure materials by recrystallization from a cooled hexane solution, which are stable in the solid-state under argon and can be stored at ambient temperatures for a few months without decomposition.

The structures of 2 and 3 were determined by multi-nuclear NMR spectroscopy and single crystal X-ray diffraction analysis; see also the SI for the details. The solid-state structures of 3a and 3b (Fig. 2b) show that their skeletal structures are very similar to each other. The $^1Au$–$^1Ge$ and $^2Au$–$^1Ge$ bond distances are 2.4475(4) and 2.4460(5) Å for 3a and 2.4371(7) and 2.4307(8) Å for 3b. On the other hand, the distance between the divalent germanium ($^2Ge$) and $^1Au$ [2.4146(4) and 2.4089(8) Å for 3a and 3b, respectively] is somewhat shorter than those of $^1Au$–$^1Ge$ and $^2Au$–$^1Ge$ bonds, suggesting a similar bonding nature between Ge(IV)–Au and Ge(II)–Au bonds. The bond angles $^1Ge$–$^1Au$–$^2Ge$ and $^1Ge$–$^2Au$–P of 3a are 168.042(5) and 175.25(3)° and those of 3b are 171.73(3) and 174.02(7)°, indicating linear arrangement of the sets of the three atoms in accord with the theoretical calculations (see below). Two germacyclopentane rings of 3a and 3b are almost perpendicular to each other with dihedral angles between the averaged ring planes of 78.915° and 84.588°, respectively. The X-ray analysis of 3a and 3b shows long $^1Au$-$^2Au$ distances of 3.913(5) and 3.9172(5)Å respectively, suggesting any aurophilic interactions to be weak at best. Significant aurophilic bonding is regarded to occur when the distance is in the range of 2.8–3.5 Å[55–60]. The $^1Au$–$^1Ge$–$^2Au$ angle of 3a (and 3b) is 106.909(16)° [and 107.61(3)°], indicating the tetrahedral geometry around $^1Ge$. The sum of the bond angles around $^2Ge$ atom of 3a (and 3b) is 359.999° (and 359.994°), which manifests the trigonal planar geometry around $^2Ge$. The unique bonding features of 3 will be discussed later on the basis of the theoretical calculations.

The $^1H$, $^{13}C$, $^{29}Si$, and $^{31}P$ NMR spectra of 3a and 3b at ambient temperatures are consistent with the structures determined by X-ray crystallography, though the spectra are complex due to the fluxionality of the molecules, whose dynamic behavior was analyzed using VT-NMR; see the SI for their detailed NMR data and spectra. In the $^1H$ NMR at room temperature in THF-$d_8$, the signals of ring and trimethylsilyl (TMS) protons of $^1GeC_4$ and $^2GeC_4$ rings of 3a (and 3b) appear at 2.28 and 0.29 ppm as broad singlets (Supplementary Fig. 1). However, at −30 °C in THF-$d_8$, three sharp singlets are observed at 0.35, 0.28, and 0.26 ppm for the TMS protons of 3b with the ratio of 2:1:1, being in accordance with the asymmetric structure with respect to the $^1GeC_4$ ring (Supplementary Fig. 27). A similar but a little more broadened

**Fig. 1 Selected examples of Au-E bond species.** Gold complexes bonded to metal and metalloid elements I–IV possessing the nucleophilic auride character.

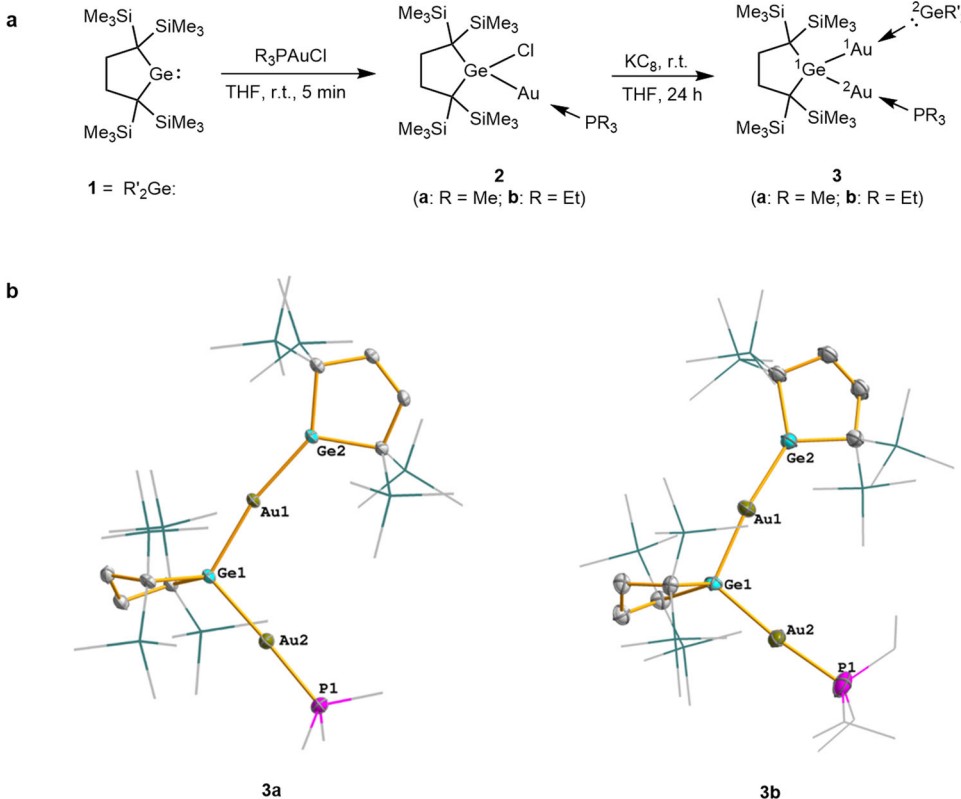

**Fig. 2 Synthesis and characterization of 3a and 3b. a** The synthesis of digoldgermanes **3a** and **3b**. **b** Molecular structures of **3a** and **3b**; hydrogen atoms are omitted for clarity. Trimethylsilyl, ethyl, and methyl groups are depicted in a wireframe model.

spectral pattern is observed in the $^1$H NMR spectrum of **3a** at −30 °C (Supplementary Fig. 22).

Broadening of the signals of TMS and ring methylene protons shown in the $^1$H NMR spectra of **3a** and **3b** suggests the fluxionality of the molecules occurring on the NMR time scale. As the methyl proton signals of PMe₃ and methyl and methylene proton signals of **3b** remain sharp even at room temperature, the dynamic process is suggested to be a pendular motion of the AuPR₃ group or the isomerization between the two equivalent structures shown in Supplementary Fig. 3. The variable temperature $^1$H NMR spectra of **3b** in the TMS proton resonance region are shown in Supplementary Fig. 2. The TMS proton signals on $^1$Ge and $^2$Ge atoms coalesce at around −10 °C. The isomerization rate $k_C$ at the coalescence temperature ($T_C$ = 263 K) is estimated as ca. 90 s⁻¹ using the equation of $k_C = \pi(v_1 - v_2)/\sqrt{2}$, where $v_1$ is the resonance frequency of TMS protons on $^2$Ge (0.35 ppm, 175 Hz) and $v_2$ as the average of two TMS resonances on $^2$Ge (0.27 ppm, 135 Hz). The activation free-energy ($\Delta G_C^{\ddagger}$) at the $T_c$ is estimated as 13.0 kcal mol⁻¹. While the mechanism of the pendular motion remains open, a plausible transition state (TS) is suggested to be **3$^T$** as shown in Supplementary Fig. 3; at the transition state, the aurophilic stabilization is supposed to be important to lower the activation energy.

The UV–Vis spectrum of **3a** in hexane shows the maximum absorption wavelength at 590 nm with the absorptivity $\varepsilon/(M^{-1}\,cm^{-1})$ of 3560 (see Supplementary Fig. 11). It is worth mentioning that the band is broad but more red-shifted than the n → 4p band of germylene **1** ($\lambda_{max}$ = 450 nm, $\varepsilon/M^{-1}\,cm^{-1}$ = 320)[61].

**Theoretical studies of 3a.** A plausible mechanism for the formation of **3** by the reduction of **2** could be proposed as shown in Supplementary Fig. 4; the initial reduction of **2** with KC₈ affords

**A** as an intermediate, and then **A** attacks another molecule of **2** in a nucleophilic manner giving **3**. Although the intermediary product **A** was not detected, the DFT calculations (at B3PW91-GD3 level in the gas phase with the basis sets of SDD level for Au) suggest that **A** would be better described as a trigonal pyramidal germyl anion as shown in Supplementary Fig. 59. The NBO analysis of **A** shows that the lone-pair electrons are largely localized on 4 s orbital of Ge with hybridization of sp$^{0.25}$ but developed to the 6p and other vacant orbitals of Au; the natural charges on Au and Ge are −0.330 and 0.502, respectively. The Au atom of **A** may serve as the nucleophilic center to attack the germanium atom of **2**.

To gain more insight into the structural characteristics of **3a** and related compounds, DFT calculations were performed at B3PW91-GD3 level (see the Methods for calculation details). As shown in Supplementary Table 2, the structural parameters of R'₂Ge(AuP)(AuGe) skeleton of **3a** determined by X-ray analysis are well reproduced by the calculations. Frontier molecular orbitals (FMOs) of **3a** are shown in Fig. 3a. HOMO and HOMO-1 are assigned as the symmetric and antisymmetric combinations of two $^1$Ge–Au σ orbitals, respectively, and LUMO has the nature of originally vacant $^2$Ge pπ orbital. The HOMO and LUMO energy levels of **3a** are −4.46 and −2.31 eV, respectively, and they are significantly higher and lower than those of germylene **1** (−5.55 and −1.77 eV at the same calculation level), suggesting higher reactivity of **3a** than **1**. The narrower HOMO-LUMO gap of **3a** (2.15 eV) than that of **1** (3.70 eV) is also in good agreement with the absorption maximum of **3a** ($\lambda_{max}$ = 590 nm) observed at longer wavelength than that of **1** ($\lambda_{max}$ = 450 nm).

Natural bond orbital (NBO) analysis of the theoretical structure of **3a** shows that it is comprised of three molecular units of R'₂GeAu₂, PMe₃, and GeR₂', where the latter two ligands coordinate to each of the two Au atoms. The $^1$Ge–$^1$Au [and

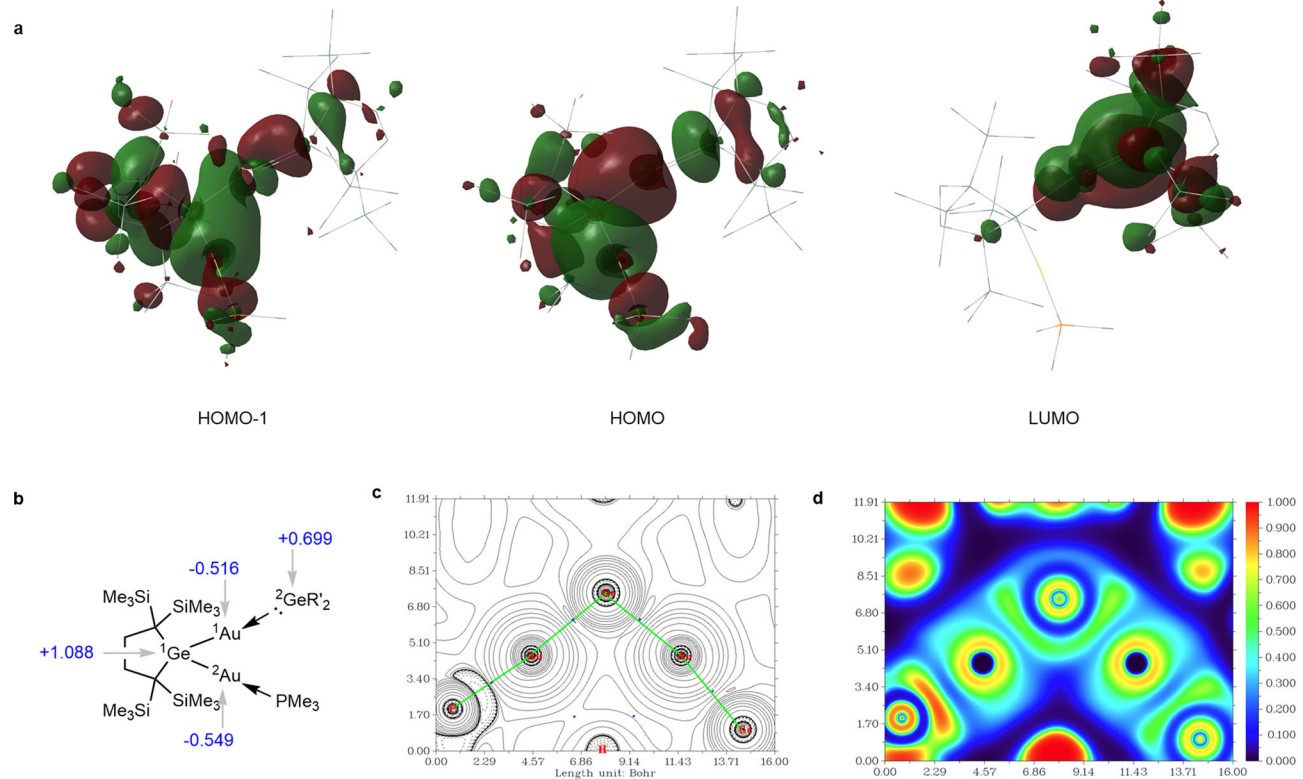

**Fig. 3 Computational studies of 3a. a** FMOs of **3a** calculated at B3PW91-GD3 level. Hydrogen atoms are omitted in the wireframe structure of **3a**.
**b** Calculated effective atomic charges of Ge and Au atoms in **3a** (the effective atomic charges of all atoms in **3a** were calculated using the AIM method).
Topological analysis for **3a**: **c** Plot of the Laplacian of the electron density on the GeAu$_2$ plane, with bond paths (light green lines), BCPs (blue dots). **d** ELF
plot on the GeAu$_2$ plane.

$^1$Ge–$^2$Au] bonds are both covalent, formed by the overlap between a $4sp^{3.13}$ hybrid orbital of $^1$Ge and 6s orbitals of $^1$Au and $^2$Au; their Wiberg bond indices are 0.6251 and 0.7122 with the occupancy of 1.8740 and 1.8909, respectively. The occupancies of the lone-pair and a vacant orbital on $^2$Ge are 1.6123 and 0.2173, showing the existence of significant dative bond between $^1$Au and $^2$Ge and small back bonding from $^1$Au to $^2$Ge. The second-order perturbation theory analysis shows the strong dative bonds from phosphine and germylene to $^2$Au and $^1$Au, respectively; the largest perturbation energy between the phosphine lone-pair orbital and $^2$Au–$^1$Ge antibonding orbital amounts to 127.5 kcal mol$^{-1}$ and that between the germylene lone-pair orbital and $^1$Au–$^1$Ge antibonding orbital is 249.8 kcal mol$^{-1}$. To study the charge distribution in the Ge–Au bond of complexes **3**, the effective atomic charges of all atoms in **3a** were calculated using AIM[62]. The effective charges of the gold atoms in **3a** are −0.516 and −0.549, while that of germanium atoms are +1.088 and +0.699 respectively (Fig. 3b). The balancing negative charge in **3a** is mostly localized at two gold atoms with the −1 oxidation state, in which gold atom demonstrates halogen-like behavior featuring two conspicuously polarized Au$^{\delta-}$–Ge$^{\delta+}$ bonds.

The quantum theory of atoms in molecules shows that there are two bond critical points (BCPs) between the $^1$Ge and two Au atoms. In addition, the other two BCPs are found between $^2$Au and P and between $^1$Au and $^2$Ge (Fig. 3c). The plot of the Laplacian of the electron density, $\nabla^2\rho(r)$, shows regions of electron density concentration on the two Au atoms (Fig. 3c). The electron localization function (ELF) plot of **3a** exhibits the inverted "V" region with the highly localized electron density at the $^1$Ge–$^1$Au and $^1$Ge–$^2$Au bonds (Fig. 3d). No BCP and

electron density concentration are found between $^1$Au and $^2$Au atoms.

**The reactivity of 3a.** Based on the DFT results, **3** could be regarded as amphoteric molecules, which involve a nucleophilic gold atom with the −1 oxidation state and a dialkylgermylene ligand bearing enhanced electrophilicity. We may expect their distinctive types of reactions. The treatment of **3a** with two moles of PMe$_3$ at room temperature (Fig. 4a) gives the corresponding digoldgermane **4**, which is coordinated by two phosphines and is isolated and characterized by X-ray (Supplementary Fig. 9) and NMR analysis (Supplementary Figs. 32–35). The HOMO of **4** holds almost the same character with that of **3a** featuring the antisymmetric combination of the two Ge–Au σ orbitals with the energy level of −4.52 eV. On the other hand, the LUMO mainly possesses the nature of the vacant Au 6p orbitals and its energy level (−0.54 eV) is much higher than that of **3a** (Supplementary Fig. 58). No reaction occurs when excess dialkylgermylene **1** is added to the solution of **4**. Neither N-heterocyclic carbene (1,3-bis(2,6-diisopropylphenyl)imidazol-2-ylidene) nor carbon monoxide (CO) reacts with **3a**.

As the negative charge in **3** is localized largely at two gold atoms, they may work as electrophiles for the addition or substitution reactions. The treatment of **3a** with methyl triflate (MeOTf), a powerful electrophilic methylation agent, at ambient temperature unexpectedly gives rise to the methylation on a germanium atom giving methylgermane **5** together with the elimination of Me$_3$PAu$^+$ moiety (Fig. 4b). The formation of Me$_3$PAuOTf is evidenced by $^{31}$P NMR ($\delta = 13.13$ ppm).

Product **5** was identified by NMR spectroscopy and single crystal X-ray diffraction analysis (Supplementary Fig. 10). The

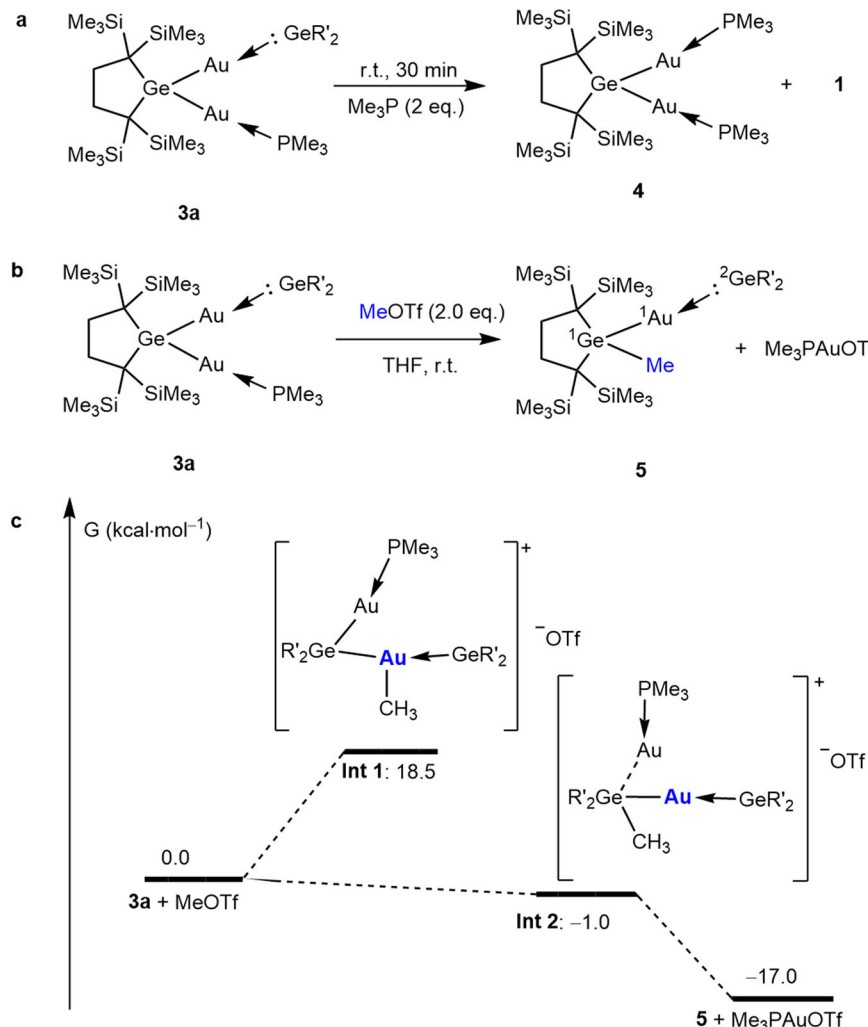

**Fig. 4 Reactivity of 3a and DFT-calculated free-energy for mechanism. a** Synthesis of **4** from digoldgermane **3a**. **b** Synthesis of **5** from digoldgermane **3a**. **c** DFT-calculated free-energy profile of a plausible mechanism for the reaction between **3a** and MeOTf in the gas phase, as determined at the B3PW91/def2-SVP level of theory.

[1]Ge–[1]Au distance [2.4330(7) Å] of **5** is similar to those of Ge–Au covalent bonds in **2** and **3** but a little longer than the bond length of [1]Au–[2]Ge [2.3997(6) Å] in **5**. The [1]Au–[1]Ge–[17]C and [1]Ge–[1]Au–[2]Ge bond angles in **5** are 105.02(18)° and 168.47(2)°, respectively. The dihedral angle between the two five-membered rings is 68.98°, smaller than that of **3a** (78.91°).

While MeOTf as an electrophile may prefer to attack an Au(–I) in **3a**, in reality, a Ge–Me bond is formed during the reaction. A pathway via the direct attack of Me[+] on a Ge atom giving **Int2** (Fig. 4c) as an intermediate is supported by the DFT calculations for two possible intermediates, **Int1** and **Int2** (Fig. 4c), which are formed by the attack of Me[+] on Au and Ge, respectively, at the B3PW91/def2-SVP level. While both intermediates are located as minima, **Int1** is found to be 19.5 kcal/mol higher in energy than **Int2** with pentacoordinate germanium atom (Fig. 4c); **Int2** is even lower in energy than that of the starting reagents **3a** + MeOTf.

Since **3a** has a low-lying LUMO, which is even lower than that of germylene **1**, the [2]Ge center of **3a** should be highly electrophilic. Facile isomerization between **3** and **3'** as shown in Supplementary Fig. 3 suggests the high electrophilicity at the [2]Ge to be attacked by an intramolecular gold nucleophile. To our anticipation, **3a** reacts readily with external nucleophiles at the [2]Ge center. The reaction of **3a** with a stoichiometric amount of

tetraphenylphosphonium chloride **6** (Ph₄P⁺Cl⁻) at ambient temperature giving the corresponding chlorogermane **7** featuring a Ge–Cl bond (Fig. 5a).

The structure of **7** was determined by NMR spectroscopy and single crystal X-ray diffraction analysis (Fig. 5b). In [1]H NMR spectra of **7** in benzene-$d_6$, trimethylsilyl proton resonances appear as four sharp singlets at 0.31, 0.25, 0.24 and 0.20 ppm, respectively, with 1:1:1:1 intensity ratio (Supplementary Fig. 39), showing that the solid-state structure is maintained in solution. A doublet [1]H NMR signal observed at 1.34 ppm is assigned to the methyl protons of a PMe₃ group. In the [31]P NMR spectrum, two signals at 26.51 and 21.20 ppm are assigned to those of a PMe₃ ligand and a Ph₄P cation, respectively (Supplementary Fig. 42). Three Ge–Au distances of **7** [2.4325(5), 2.4284(5), and 2.4458(5)Å] are similar to those of Ge–Au covalent bonds found for **2** and **3** but a little longer than the [1]Au–[2]Ge distance of **3a**. The [1]Ge–[2]Au–P and [1]Ge–[1]Au–[2]Ge bond angles of **7** are 173.81(4)° and 163.649(17)°, respectively. The dihedral angle between the two five-membered rings is 78.915°, which is similar with that found in **3a**.

The reaction of **3a** with acetyl chloride **8** at ambient temperature afforded chlorogoldgermane **9** in 67%, while an expected by-product, acetyl(trimethylphosphine)gold, is not detected (Fig. 5a). The reaction of **3a** with **8** may produce **7** anion with an acetyl counterion, but in reality gives the

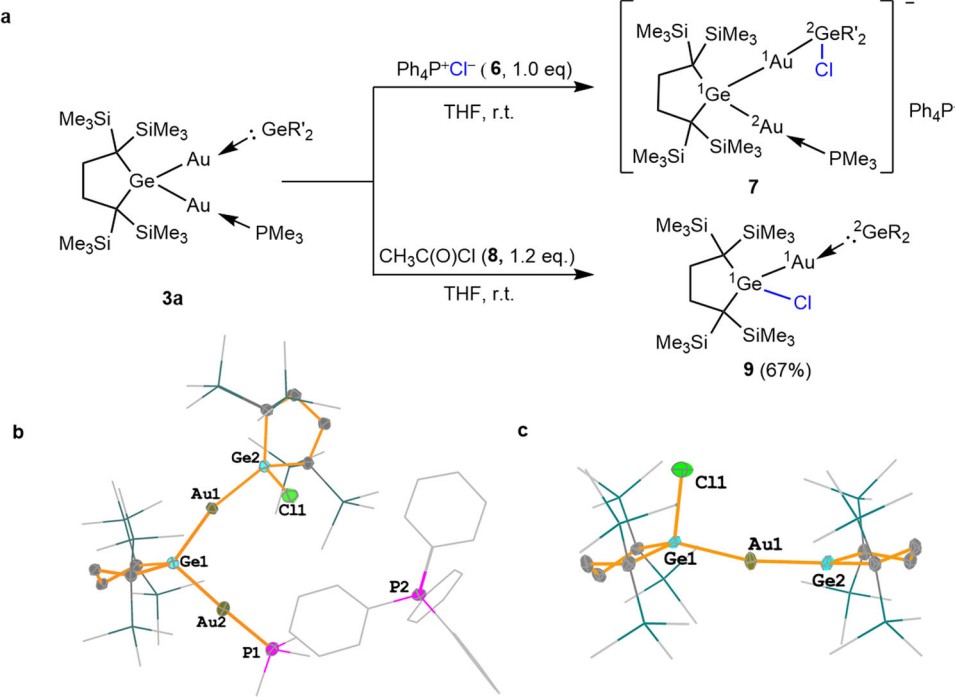

**Fig. 5 Reactivity of 3a and structural characterization of products. a** The reactions of **3a** with tetraphenylphosphonium chloride (Ph₄PCl) **6**, and acetyl chloride. **b** Molecular structure of **7** (Hydrogen atoms are omitted for clarity. Trimethylsilyl, ethyl, phenyl and methyl groups are depicted in a wireframe model). **c** Molecular structure of **9** (Hydrogen atoms are omitted for clarity. Trimethylsilyl, ethyl, phenyl and methyl groups are depicted in a wireframe model).

corresponding chlorogermane **9**. As acetyl cation is more electrophilic than $Ph_4P^+$, the $^1Ge–^2Au$ bond of **7** once formed would be cleaved by the acetyl counter cation to generate **9**; the formation of Ge–Cl bond and the cleavage of $^1Ge–^2Au$ bond may occur concertedly as shown in Supplementary Fig. 6. The structure of **9** was determined by NMR spectroscopy and single crystal X-ray diffraction analysis (Fig. 5c). The $^1Au–^1Ge–Cl$ and $^1Ge–^1Au–^2Ge$ bond angles of **9** are 97.614(65)° and 167.659(34)°, respectively. Two germacyclopentane rings of **9** are almost in-plane with the dihedral angle of 166.99°, which is very different from those of **3**, **5**, and **7** (69–85°).

**The catalysis reactivity of 3a.** Digoldgermane **3a** has been found to exhibit effective catalytic ability for the cyclic trimerization of aryl isocyanates (Supplementary Fig. 7). In the presence of 0.01 mol% of **3a**, the trimerization of various phenyl isocyanates **10a–e** takes place smoothly giving triaryl isocyanurates **11a–e** in 78–98% isolated yields (Supplementary Fig. 7). Neither related goldgermanes **2a**, **4a** nor dialkylgermylene **1** show the catalytic activity for the trimerization of aryl isocyanates. There have been many catalysts discovered and diverse mechanisms have been proposed for the trimerization[63–65]. It is inferred that the high electrophilicity of the $^2Ge$ atom in **3a** is essential for the catalytic activity; see Fig. S3 for a proposed catalytic cycle.

## Discussion

In conclusion, digoldgermanes **3** with a germylene ligand have been synthesized through the reaction of stable dialkylgermylene **1** with $(R_3P)AuCl$ followed by the $KC_8$ reduction. The DFT calculations of **3a** show that the HOMO is high-lying with the nature of Ge–Au σ bonding orbital and the LUMO has largely germylene vacant 4pπ nature with significantly lower energy level than that of **1**. Digoldgermanes **3** feature amphiphilic reactivity with the electrophilic Ge(II) atom and the nucleophilic Au(-I)–Ge(IV)

bond. Digoldgermanes **3** show: (1) the pendular motion of $AuPR_3$ ligand between two germanium atoms of **3** occurring on the NMR time scale; (2) the electrophilic methylation of **3a** with MeOTf occurring at the Ge(IV) atom rather than the Au(-I) atom; (3) the facile reactions with the nucleophiles, $Ph_4P^+Cl^-$ and acetyl chloride, giving the chlorination products, while the latter is accompanied by the Ge(IV)–Au bond cleavage; and (4) the catalytic activity towards the cyclic trimerization of aryl isocyanates giving the corresponding triaryl isocyanurates.

## Methods

**General synthetic procedure**. All reactions were performed under an atmosphere of argon by using standard Schlenk or dry box techniques; solvents were dried over Na metal or $CaH_2$ under nitrogen atmosphere. $(R_3P)AuCl$ (R = Me, Et) were synthesized using literature procedures[66–68]. $^1H$, $^{13}C$, $^{29}Si$, and $^{31}P$ NMR spectra were obtained with a Bruker AV 400 instrument at 400 MHz ($^1H$ NMR), 101 MHz ($^{13}C$ NMR) and 162 MHz ($^{31}P$ NMR), as well as Bruker AV 500 instrument at 500 MHz ($^1H$ NMR), 126 MHz ($^{13}C$ NMR), 99 MHz ($^{29}Si$ NMR), 202 MHz ($^{31}P$ NMR) at 298 K. Unless otherwise noted, the NMR spectra were recorded in benzene-$d_6$ at ambient temperature. The $^1H$ and $^{13}C$ NMR chemical shifts were referenced to residual $^1H$ and $^{13}C$ signals of the solvents. NMR multiplicities are abbreviated as follows: $s$ = singlet, $d$ = doublet, $t$ = triplet, $dt$ = doublet of triplets, $m$ = multiplet, and $brs$ = broad singlet. Coupling constants $J$ are given in Hz. Electrospray ionization (ESI) mass spectra were obtained at the Mass Spectrometry Laboratory at Hangzhou Normal University with a Bruker Daltonics MicroQtof spectrometer. Melting points were measured with a BUCHI Melting Point M-560. Sampling of air-sensitive compounds was carried out using a MBRAUN's MB-10-G glove box. UV–Vis spectra were recorded on a Shimadzu UV-1800 spectrophotometer.

**Synthesis of gold(I) complexes 2a and 2b**. In a glove box, $(R_3P)AuCl$ (R = Me or Et, 1.0 mmol) was added into a THF (10 mL) solution of germylene **1** (350 mg, 1.02 mmol) and the mixture was stirred at room temperature for 5 min. The solvent was removed under vacuum to afford the residue, which was washed with hexane (5 mL) for 3 times. The residual solvents were evaporated in vacuo affording gold(I) complex **2a** or **2b**, which were stable under argon for a few months. **2a** (547 mg, 90%): white powder. **M.p.**: 153 °C (dec.); $^1H$ NMR (400 MHz, $C_6D_6$, 25 °C) δ 2.46–2.38 (m, ring-$CH_2$, 2H), 2.20–2.12 (m, ring-$CH_2$, 2H), 0.60 (s, $SiCH_3$, 18H), 0.51 (s, $SiCH_3$, 18H), 0.46 (d, $^2J_{H-P}$ = 9.20 Hz, $PCH_3$, 9H); $^{13}C$ NMR (101 MHz, $C_6D_6$, 25 °C) δ 35.28 (d, $^4J_{C-P}$ = 2.93 Hz, ring-$CH_2$), 26.60 (d, $^3J_{C-P}$ = 14.04 Hz, ring-$C^q$), 14.87 (d, $^1J_{C-P}$ = 28.58 Hz, $PCH_3$), 5.23 ($SiCH_3$), 5.13

(SiCH$_3$); **$^{29}$Si NMR** (99 MHz, C$_6$D$_6$, 25 °C) $\delta$ 4.17, 2.02; **$^{31}$P NMR** (162 MHz, C$_6$D$_6$, 25 °C) $\delta$ 26.01; **HRMS** (ESI): $m/z$ calcd. for C$_{19}$H$_{49}$AuGeClPSi$_4$ (M$^+$): 725.9677, $m/z$ calcd. for [M–$Cl$]$^+$, 691.1538, found: 691.1524. **2b** (580 mg, 92%): white powder. **M.p.**: 172 °C (dec.); **$^1$H NMR** (400 MHz, C$_6$D$_6$, 25 °C) $\delta$ 2.45–2.39 (m, ring-CH$_2$, 2H), 2.18–2.13 (m, ring-CH$_2$, 2H), 0.96–0.88 (m, PCH$_2$, 6H), 0.71 (dt, PCH$_2$CH$_3$, $^3J_{H-P}$ = 17.60 Hz, $^3J_{H-H}$ = 7.60 Hz, 9H), 0.61 (s, SiCH$_3$, 18H), 0.51 (s, SiCH$_3$, 18H); **$^{13}$C NMR** (101 MHz, C$_6$D$_6$, 25 °C) $\delta$ 35.29 (d, $^4J_{C-P}$ = 2.52 Hz, ring-CH$_2$), 26.56 (d, $^3J_{C-P}$ = 13.13 Hz, ring-C$^q$), 17.97 (d, $^1J_{C-P}$ = 26.06 Hz, PCH$_2$), 8.76 (s, PCH$_2$CH$_3$), 5.25 (s, SiCH$_3$), 5.06 (s, SiCH$_3$); **$^{29}$Si NMR** (99 MHz, C$_6$D$_6$, 25 °C) $\delta$ 4.10, 2.14; **$^{31}$P NMR** (162 MHz, C$_6$D$_6$, 25 °C) $\delta$ 60.48. **HRMS** (ESI): $m/z$ calcd. for C$_{22}$H$_{55}$AuGeClPSi$_4$(M$^+$) 768.0474, $m/z$ calcd. for [M–$Cl$]$^+$ 733.1994.

**Synthesis of gold complexes 3a and 3b.** 2a or 2b (1.00 mmol) was mixed with KC$_8$ (1.05 mmol, 177 mg) in THF (10 mL). The mixture was stirred at ambient temperature for 12 h. Then the mixture was concentrated under vacuum. The reside was washed with hexane (4 mL) for three times and extracted with toluene (20 mL). The extracts were concentrated under vacuum to afford **3a** or **3b** as blue-green solids. **3a** (347 mg, 49%): **M.p.**: 150 °C (dec.); **$^1$H NMR** (400 MHz, C$_6$D$_6$, 25 °C) $\delta$ 2.33 (brs, ring-CH$_2$, 8H), 0.89 (d, $^2J_{H-P}$ = 7.60 Hz, PCH$_3$, 9H), 0.49 (brs, SiCH$_3$, 72H); **$^1$H NMR** (600 MHz, THF-$d_8$, –30 °C) $\delta$ 2.47 (brs, ring-CH$_2$, 4H), 2.07 (brs, ring-CH$_2$, 4H), 1.43 (d, $^2J_{H-P}$ = 8.20 Hz, PCH$_3$, 9H), 0.33 (brs, SiCH$_3$, 36H), 0.25 (s, SiCH$_3$, 18H), 0.22 (s, SiCH$_3$, 18H); **$^{13}$C NMR** (101 MHz, C$_6$D$_6$, 25 °C) $\delta$ 36.99 (ring-CH$_2$), 30.22 (ring-C$^q$), 16.20 (d, $^1J_{C-P}$ = 21.61 Hz, PCH$_3$), 4.36 (brs, SiCH$_3$); **$^{29}$Si NMR** (99 MHz, C$_6$D$_6$, 25 °C) $\delta$ 0.14 (brs); **$^{31}$P NMR** (202 MHz, C$_6$D$_6$, 25 °C) $\delta$ 39.10; **UV/Vis**: $\lambda_{max}$ 590 nm; **HRMS** (ESI): $m/z$ calcd. for [C$_{32}$H$_{80}$AuGe$_2$Si$_8$]$^-$: 1032.2535, found: 1032.2587. **3b** (380 mg, 54%): **M.p.**: 162 °C (dec.); **$^1$H NMR** (500 MHz, THF-$d_8$, 25 °C) $\delta$ 2.28 (brs, ring-CH$_2$, 8H), 1.81 (m, PCH$_2$, 6H), 1.22 (m, PCH$_3$, 9H), 0.29 (brs, SiCH$_3$, 72H); **$^1$H NMR** (600 MHz, THF-$d_8$, –30 °C) $\delta$ 2.47 (brs, ring-CH$_2$, 4H), 2.07 (brs, ring-CH$_2$, 4H), 1.82 (p, $^2J_{H-P}$ = 7.7 Hz, 6H), 1.21 (dt, $^3J_{H-P}$ = 16.9, $^3J_{H-H}$ = 7.6 Hz, 9H), 0.32 (brs, SiCH$_3$, 36H), 0.25 (s, SiCH$_3$, 18H), 0.23 (s, SiCH$_3$, 18H); **$^{13}$C NMR** (126 MHz, THF-$d_8$, 25 °C) $\delta$ 37.20 (s, ring-CH$_2$), 19.17 (d, $^1J_{C-P}$ = 20.50 Hz, PCH$_2$), 8.84 (d, $^2J_{C-P}$ = 1.52 Hz, PCH$_2$CH$_3$), 4.26 (s, SiCH$_3$); **$^{29}$Si NMR** (99 MHz, C$_6$D$_6$, 25 °C) $\delta$ 0.13 (brs); **$^{31}$P NMR** (202 MHz, THF-$d_8$, 25 °C) $\delta$ 55.10; **UV/Vis**: $\lambda_{max}$ 596 nm; **HRMS** (ESI): $m/z$ calcd. for [C$_{32}$H$_{80}$AuGe$_2$Si$_8$]$^-$: 1032.2535, found: 1032.2597.

**Synthesis of complex 4.** PMe$_3$ (12 mg, 0.16 mmol) was added into a THF (2 mL) solution of **3a** (100 mg, 0.08 mmol) and the mixture was stirred at room temperature for 20 min. The solution color changed from blue to light green. The reside was washed with cooled hexane (2 mL) for three times and extracted with toluene (2 mL). The extracts were concentrated under vacuum to give **4** as light green crystals in 92% yield (71 mg): **M.p.**: 148 °C (dec.); **$^1$H NMR** (400 MHz, C$_6$D$_6$, 25 °C) $\delta$ 2.45 (s, ring-CH$_2$,4H), 0.7 (s, SiCH$_3$, 36H), 0.68 (d, PCH$_3$, $J_{H-P}$ = 7.6 Hz, 18H); **$^{13}$C NMR** (126 MHz, C$_6$D$_6$, 25 °C) $\delta$ 37.42 (ring-CH$_2$), 19.74 (t, $^3J_{C-P}$ = 6.0 Hz, ring-C$^q$), 15.98 (dd, $^1J_{C-P}$ = 21.4 Hz, $^5J_{C-P}$ = 3.8 Hz, PCH$_3$), 5.86 (SiCH$_3$); **$^{29}$Si NMR** (99 MHz, C$_6$D$_6$, 25 °C) $\delta$ 1.93; **$^{31}$P NMR** (162 MHz, C$_6$D$_6$, 25 °C) $\delta$ 40.95; **HRMS** (ESI): $m/z$ calcd. for [C$_{22}$H$_{58}$Au$_2$GeP$_2$Si$_4$]$^+$: 964.1624, found: 964.1632.

**Synthesis of complex 5.** In a glove box, MeOTf (25.1 mg, 0.153 mmol) was added into a THF (10 mL) solution of **3a** (100 mg, 0.0766 mmol) and the mixture was stirred at room temperature for 24 h. The solvent was removed under vacuum to afford the residue, which was extracted with hexane (20 mL). After evaporation of the solvents in vacuo, recrystallization from hexane gave pure **5** (45.9 mg, 57%): red solids; **M.p.**: 188 °C; **$^1$H NMR** (400 MHz, C$_6$D$_6$, 25 °C) $\delta$ 2.22 (m, ring-CH$_2$, 4H), 2.16 (s, ring-CH$_2$, 4H), 1.11 (s, CH$_3$, 3H), 0.56 (s, SiCH$_3$, 18H), 0.43 (s, SiCH$_3$, 18H), 0.23 (s, SiCH$_3$, 36H); **$^{13}$C NMR** (101 MHz, C$_6$D$_6$, 25 °C) $\delta$ 62.09 (ring-C$^q$), 36.58 (ring-CH$_2$), 35.92 (ring-CH$_2$), 18.90 (ring-C$^q$), 11.31 (CH$_3$), 5.55 (SiCH$_3$), 5.21(SiCH$_3$), 2.69 (SiCH$_3$); **$^{29}$Si NMR** (99 MHz, C$_6$D$_6$, 25 °C) $\delta$ 3.41, 2.43, 1.32. **HRMS** (ESI): $m/z$ calcd. for [M + $Cl$]$^-$: [C$_{33}$H$_{83}$AuGe$_2$Si$_8$Cl]$^-$ 1083.2416, found: 1083.2439.

**Synthesis of complex 7.** In a glove box, PPh$_4$Cl (29.4 mg, 0.078 mmol) was added into a THF (10 mL) solution of **3a** (100 mg, 0.0766 mmol) and the mixture was stirred at room temperature for 24 h. The solvent was removed under vacuum to afford the residue that was washed with ether (5 mL) for three times. Finally, compound **7** was obtained as light-red powder. **7** (125 mg, 97%): **M.p.**: 172 °C (dec.); **$^1$H NMR** (400 MHz, THF-$d_8$, 25 °C) $\delta$ 7.97-7.93 (t, Ar-H, $J$ = 7.2 Hz, 4H), 7.79-7.75 (m, Ar-H, 16H), 2.25-2.23 (t, ring-CH$_2$, $J$ = 5.60 Hz, 2H), 2.09-2.07 (t, ring-CH$_2$, $J$ = 6.4 Hz, 2H), 2.00 (s, ring-CH$_2$, 4H), 1.35-1.33 (d, PCH$_3$, $^2J_{P-H}$ = 7.6 Hz, 9H), 0.31 (s, SiCH$_3$, 18H), 0.25 (s, SiCH$_3$,18H), 0.23 (s, SiCH$_3$, 18H), 0.2 (s, SiCH$_3$, 18H); **$^{13}$C NMR** (126 MHz, THF-$d_8$, 25 °C) $\delta$ 136.28 (Ar-C), 135.48 (d, Ar-C, $^3J_{P-C}$ = 11.0 Hz), 131.20 (d, Ar-C, $^2J_{P-C}$ = 11.1 Hz), 118.94 (d, $^1J_{P-C}$ = 89.7 Hz), 37.36 (ring-CH$_2$), 36.21(ring-CH$_2$), 28.82 (ring-C$^q$), 18.96 (d, PCH$_3$, $J$ = 5.4 Hz), 16.82 (d, ring-C$^q$, $J$ = 18.9 Hz), 5.77 (SiCH$_3$), 5.46 (SiCH$_3$); **$^{29}$Si NMR** (99 MHz, THF-$d_8$, 25 °C) $\delta$ 1.81, 1.58, 1.28, −0.04; **$^{31}$P NMR** (162 MHz, THF-$d_8$, 25 °C) $\delta$ 26.47 (PMe$_3$), 21.21 (PPh$_4$); **HRMS** (ESI): $m/z$ calcd. for [C$_{35}$H$_{89}$Au$_2$ClGe$_2$PSi$_8$]$^-$: 1341.2299, found: 1341.2312.

**Synthesis of complex 9.** In a glove box, CH$_3$COCl (18.0 mg, 0.23 mmol) was added into a benzene (10 mL) solution of **3a** (150 mg, 0.115 mmol) and the mixture was stirred at room temperature for 10 h. The solvent was removed under vacuum to afford the residue that was extracted with hexane (5 mL) for three times. Finally, compound **9** was obtained as rufous solid. **9** (125 mg, 67%): **M.p.**: 172 °C (dec.); **$^1$H NMR** (400 MHz, C$_6$D$_6$, 25 °C) $\delta$ 2.49-2.45 (m, ring-CH$_2$, 2H), 2.26-2.22 (m, ring-CH$_2$, 2H), 2.13 (s, ring-CH$_2$, 4H), 0.56 (s, SiCH$_3$, 18H), 0.49 (s, SiCH$_3$, 18H), 0.20 (s, SiCH$_3$, 36H); **$^{13}$C NMR** (126 MHz, C$_6$D$_6$, 25 °C) $\delta$ 36.44 (ring-CH$_2$), 35.21 (ring-CH$_2$), 27.26 (ring-CH$_2$), 27.19 (ring-CH$_2$), 4.80 (SiCH$_3$), 4.78 (SiCH$_3$), 4.43 (SiCH$_3$), 4.40 (SiCH$_3$), 2.66 (SiCH$_3$), 2.62 (SiCH$_3$); **$^{29}$Si NMR** (99 MHz, C$_6$D$_6$, 25 °C) $\delta$ 3.55, 3.45, 0.67, 0.57, −1.02; **HRMS** (ESI): $m/z$ calcd. for [M–$Cl$]$^+$: [C$_{32}$H$_{80}$AuGe$_2$Si$_8$]$^+$ 1083.2416, found: 1083.2439.

**General procedure for cyclic trimerization of aryl isocyanates catalyzed by 3a.** A THF solution of complex **3a** (7.66 × 10$^{-4}$ M in THF, 130 μL, 0.01 mol%) was introduced in a thick-walled tube, which contained THF (2.0 mL) and aryliso-cyanate **10** (1 mmol). The reaction mixture was heated at 80 °C for 14 h. After being washed with $n$-hexane, the corresponding triarylisocyanurate **11** was obtained. **11a** (Ar = Ph, 116 mg, 97%): **$^1$H NMR** (400 MHz, CDCl$_3$, 25 °C) $\delta$ 7.53-7.45 (m, Ar-H, 6H), 7.43-7.40 (m, Ar-H, 6H); **$^{13}$C NMR** (101 MHz, CDCl$_3$, 25 °C) $\delta$ 148.84 (O = C), 133.73 (Ar-C), 129.51 (Ar-C), 128.55 (Ar-C). **11b** (Ar = $p$-ClC$_6$H$_4$, 132 mg, 86%): **$^1$H NMR** (400 MHz, CDCl$_3$, 25 °C) $\delta$ 7.49-7.47 (d, $J$ = 8.8 Hz, Ar-H, 6H), 7.33-7.31 (d, $J$ = 8.8 Hz, Ar-H, 6H); **$^{13}$C NMR** (101 MHz, CDCl$_3$, 25 °C) $\delta$ 148.27 (O = C), 135.73 (Ar-C), 131.86 (Ar-C), 129.86 (Ar-C), 129.86 (Ar-C). **11c** (Ar = $p$-CF$_3$C$_6$H$_4$, 135 mg, 72%): **$^1$H NMR** (400 MHz, C$_6$D$_6$, 25 °C) $\delta$ 7.81-7.79 (d, $J$ = 8.4 Hz, Ar-H, 6H), 7.56-7.54 (d, $J$ = 8.4 Hz, Ar-H, 6H); **$^{13}$C NMR** (101 MHz, CDCl$_3$, 25 °C) $\delta$ 147.92 (O = C), 136.27 (Ar-C), 132.00 (q, $^2J_{C-F}$ = 33.13, Ar-C), 129.23 (Ar-C), 126.83 (q, $^3J_{C-F}$ = 3.63, Ar-C), 122.27 (C-F). **11d** (Ar = $p$-CH$_3$C$_6$H$_4$, 131 mg, 98%): **$^1$H NMR** (400 MHz, CDCl$_3$, 25 °C) $\delta$ 7.26 (d, $J$ = 2.4 Hz, Ar-H, 12H), 2.38 (s, CH$_3$, 9H); **$^{13}$C NMR** (101 MHz, CDCl$_3$, 25 °C) $\delta$ 148.98 (O = C), 139.35 (Ar-C), 131.20 (Ar-C), 130.07 (Ar-C), 128.18 (Ar-C), 21.32 (CH$_3$). **11e** (Ar = $p$-MeOC$_6$H$_4$, 147 mg, 98%): **$^1$H NMR** (400 MHz, CDCl$_3$, 25 °C) $\delta$ 7.30-7.28 (d, $J$ = 8.8 Hz, Ar-H, 6H), 6.99-6.97 (d, $J$ = 8.8 Hz, Ar-H, 6H), 3.82 (s, OCH$_3$, 9H); **$^{13}$C NMR** (101 MHz, CDCl$_3$, 25 °C) $\delta$ 159.99 (Ar-C), 149.24(O = C), 129.52 (Ar-C), 126.39 (Ar-C), 114.69 (Ar-C), 55.59 (OCH$_3$).

**Theoretical calculations.** Theoretical calculations were performed for the compounds **1, 3a, 4**, and anion **A**, using the Gaussian 16 program package[69] at TianHe-2 located in Shanxi Supercomputing Center. The structures phase were optimized using a dispersion-corrected DFT method at the B3PW91[70]-GD3 level[71,72] with the basis sets of 6-31 + G(d,p) for C, H, Si, P, and Ge atoms + SSD for Au. As shown in Table S2, the structural parameters of R'$_2$Ge(AuP)(AuGe) moiety of **3a** determined by X-ray analysis were well reproduced by the calculations. The structures related to the reaction of **3a** with MeOTf were optimized at the B3PW91/def2-SVP[73] level in the gas phase. All of the structures obtained herein were verified by examination of their Hessian matrix as minima (all frequencies real). The solvent effects on the relative stability of the compounds were not evaluated. The AIM charges of the atoms were calculated using the basin analysis module of Multiwfn3.8[62].

## Data availability

Metrical data for the solid-state structures of **2b, 3a, 3b, 4, 5, 7**, and **9** in this paper have been deposited at the Cambridge Crystallographic Data Centre under reference numbers CCDC: 2040233, 2040232, 2040235, 2040234, 2093574, 2040236, and 2093575, respectively. Copies of the data can be obtained free of charges from www.ccdc.cam.ac.uk/structures/. All other data supporting the findings of this study are available within the article and its Supplementary Information.

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

## Acknowledgements

This work was financially supported by the National Natural Science Foundation of China (Grant No. 22071039, and 22101068), and the Natural Science Foundation of Zhejiang Province (Grant No. LY19B020007 and LQ21B020007). We are extremely grateful to Profs. Liejin Zhou and Lichun Kong of Zhejiang Normal University for the contributions of variable temperature (VT) NMR experiments.

## Author contributions

L.W. performed preliminary experiments on the system. G.Z., Y.L., L.Y., L.H., and X.C. carried out the synthetic work and analytical characterization. L.W. and M.K. performed DFT calculations. X.C. acquired the XRD data. L.W., X.C., M.K., and Z.L. wrote the paper, all authors discussed and commented on the manuscript. M.K. and Z.L. directed and coordinated the research.

## Competing interests

The authors declare no competing interests.
