## [Peer Review File · Nature Communications]

Unique Digoldgermanes Involving Structural Characteristics,
Dynamic Behaviour and Distinctive ReactionsREVIEWER COMMENTS

Reviewer #1 (Remarks to the Author):

The contribution by Zhifang Li et al. describes the synthesis and full characterization of germylene-bridged digold complexes. This is a quite remarkable finding as digold complexes with a monoatomic bridge are very rare indeed. However, it would be of paramount importance to discuss possible aurophilic interactions in more detail. The Au-Au distance in 3a,b is not given, but is probably at about 3.8 Å, which would be rather long for aurophilic interactions. Nonetheless, the canonical orbitals suggest a small degree of constructive interaction.

The manuscript is unfortunately quite difficult to read. The number of language related issues is too high to provide a comprehensive list. There is insufficient context to really appreciate the chemistry of polynuclear gold complexes and clusters. The discussion of the DFT results remains a bit superficial. Prior to acceptance, the whole manuscript needs to be edited to acceptable standards.

As noted before, this reviewer feels unable to provide a comprehensive list of necessary changes.

P1, L11: "...with gold centres coordinated by dialkylgermylene ligands,..."

P1, 16: The pendular motion is a plausible explanation for the observations in VT NMR. It was hardly "found" by VT NMR, which does not provide unambiguous evidence.

P1, 19: What does "closes to -1 oxidation state" mean?

P1, 20: (1) The sentence starting with "We show the bond formation..." does not make any sense. Methylgermane 5 only contains a methyl group at germanium.

P1, 26: The interest in catalysis, microelectronics, materials and pharmacology is anything but new. What is the point of this claim anyway?

P2, 29: The "chi" symbol for electronegativity is displayed as a box in the pdf-file.

P2, 35: Delete "firstly"

P2, 38: Sentence is nonsense: the lower electronegativity of boron may result in Au-B bonds with boron auride character. It is, however, no prerequisite for Au-B formation at all!

P3, 46: "intermediacy of V during these reactions was suggested."

P3, 47: Why "particularly interesting"? All p-block metals and semi metals are less electronegative than gold!

P3, 52: "..., isolable digoldgermanes are still unknown to date." While this statement is probably correct, it would be appropriate to discuss known species with two gold atoms bridged by a single atom. For instance, μ -imido complexes without obvious aurophilic interactions have been reported by Sharp et al.

P3, 64: "as white solids"

P3, 65: Delete superfluous "until now".

P4, 69: "can be stored at ambient temperature for a few months [without decomposition]."

P4, 72: Replace "like each other" by "very similar"

P4, 75: How does a SLIGHT [bare significant] shortening suggest SIGNIFICANT differences in bonding?

P4, 79: A bond angle of close to 180° does not indicate almost-linearity, it is a manifestation of almost-linearity.

P4, 80: Why invoke VSEPR? Are any lone-pairs involved?

P4, 80: Why INTRINSICALLY perpendicular? They are close to perpendicular, that's all. This is presumably due to steric reasons as 3 is not an allene after all.

P4, 84: See comments before. A sum of angles of close to 360° is a manifestation of a

planar coordination environment, not evidence.

P4, 85: Only one unique bonding feature?

P5, 96: What is DNMR? Do the authors mean VT-NMR?

P6, 112: Replace evaluated by "estimated". The equation is a relatively rough approximation.

P6, 117: The sentence is very similar to the previous sentence. Please reword to avoid repetition.

P7, 125: What does the sentence starting with "Au center" mean?

Reviewer #2 (Remarks to the Author):

The manuscript titled: Unique Digoldgermanes: Structural Characteristics, Dynamic Behaviour and Distinctive Reactions by Wang, Zhen, Li, Kira, Yan, Chang, Huang, and Li can be considered a nice extension of the recent paper by the same authors in *Angew. Chem.* on digoldstannanes.

In the paper, the authors describe the preparation of an interesting class of compounds where Kira's germylene is coordinated to two gold atoms. One of these gold atoms bears a phosphine ligand, whereas the second one is coordinated by another germylene. The presence of two germylenes and two gold atoms in the compound allows for an interesting dynamic behavior, where the two germylenes switch roles. This "pendulum motion" is nicely studied by VT NMR spectroscopy and also by theoretical calculations. Furthermore, the reactivity of the compound toward nucleophiles and electrophiles was investigated to some extent. Finally, the compound was found to exhibit remarkable catalytic properties for the trimerization of arylcyanates.

In the experimental part, the reaction of germylene 1 with Ph₄PCl is described. I did not find an explanation for this in the text and in addition, I have no idea what the source of the proton in the final product is!

I found a number of things that require attention as listed below. Nothing too serious!

- In line 74 replace "a divalent" by "the divalent".
- In the sentence starting in line 74 it is stated that "the distance between the divalent germanium (2Ge) and 1Au [2.4146(4) and 2.4089(8) Å for 3a and 3b, respectively] is slightly shorter than those of 1Au-1Ge and 2Au-1Ge bonds, suggesting significant difference in the bonding nature between Ge(IV)-Au and Ge(II)-Au bonds." If there is only a slight effect, how can this suggest a significant difference??
- In line 95 the spectra are a little confused. I suppose what the authors rather mean is confusing.
- In scheme 2 I wonder if it might be more appropriate to draw the curved arrow of 3 and 3' between the two gold atoms because this interaction leads to 3T. In addition in 3T curved arrows might be drawn to the two Ge atoms.
- The next sentence speculates about back bonding between Ge and Au. However, the calculations later disprove this. So why not omit this sentence?
- On page 8 the reference to the figure containing effective charges should be to Figure 4 and not 3 (line 158).
- The authors own ref 42 Ref. is more likely "Kira, M. et al." than "Mitsuo, K. et al."
- On page 9 it is stated that "no reaction occurs when excess stannylene is added in the solution of 3a." What "stannylenes" do the authors refer to here (line 172)?
- In line 173 replace "N-Heterocyclic" by "N-heterocyclic"
- In line 175 make compound number 3 bold and replace "tow" by "two".
- In line 176 put a period after "reactions".
- In line 182 replace Fig. 4 by Fig. 5. Is "Three Ge1-Au1 distance" supposed to mean "The

Ge1-Au1 distance"??

- In the caption of Fig. 5 replace "structures" by "structure".
- In Fig. 7 incorporate compound number 8 for acetyl chloride.
- In line 262 "replace "13C" by "13 signals".

My overall impression of the manuscript is very positive. Interesting chemistry, carried very well. Thorough discussion and excellent execution. Quality of spectra and crystal structures is good.

However, what I miss in this paper is the "wow"-factor. The feeling that this is something completely unexpected or new, something I would not have thought about myself. This arguably subjective criterion would justify publication in Nature Communications. But my personal impression is not "wow" but "interesting, they are doing the things they have done with tin now also with germanium."

For this reason, I do not support publication of this really nice manuscript in in Nature Communications but suggest publishing it elsewhere.

We deeply appreciate the two reviewers for their very careful and detailed comments. These are very valuable for us to revise the manuscript. The manuscript has been revised in light of the reviewers' comments, where the sentences and paragraphs highlighted with yellow color are our responses to the comments of reviewers.

Reviewer #1 (Remarks to the Author):

1. The contribution by Zhifang Li et al. describes the synthesis and full characterization of germylene-bridged digold complexes. This is a quite remarkable finding as digold complexes with a monoatomic bridge are very rare indeed. However, it would be of paramount importance to discuss possible aurophilic interactions in more detail. The Au-Au distance in **3a,b** is not given, but is probably at about 3.8 Å, which would be rather long for aurophilic interactions. Nonetheless, the canonical orbitals suggest a small degree of constructive interaction.

Response: *We are also interested in the possible aurophilicity in **3**. However the $^1\text{Au}-^2\text{Au}$ distances for **3a** and **3b** are longer than 3.9 Å, and hence, we concluded there is no significant aurophilic interaction between the two Au atoms in the molecules. This point is discussed in the 2nd paragraph of p. 3. The aurophilic interaction may be important in the pendular motion. As discussed in the middle of p. 4, at the transition state **3^T** shown in Scheme 2, the aurophilic stabilization is supposed to be important to lower the activation energy.*

2. The manuscript is unfortunately quite difficult to read. The number of language related issues is too high to provide a comprehensive list. There is insufficient context to really appreciate the chemistry of polynuclear gold complexes and clusters.

Response: *We have checked the whole text and made efforts to modify, add or delete appropriate sentences so as to express straightforwardly what we mean. Sentences inserted, and modified were shown by highlight with yellow color.*

3. The discussion of the DFT results remains a bit superficial.

Response: *The results of AIM calculations were discussed in p.6. We added the plot of the*

Laplacian of the electron density on the GeAu₂ plane, with bond paths, BCPs, NNA and ELF plot on the GeAu₂ plane in Fig 5 (p.7).

4. Prior to acceptance, the whole manuscript needs to be edited to acceptable standards. As note before, this reviewer feels unable to provide a comprehensive list of necessary changes. P1, L11: "...with gold centres coordinated by dialkylgermylene ligands..."

Response: *The sentence was modified as "Unique digold-substituted germanes of the form of R'₂Ge(AuPR₃)(AuGeR'₂) -----."*

5. P1, 16: The pendular motion is a plausible explanation for the observations in VT NMR. It was hardly "found" by VT NMR, which does not provide unambiguous evidence.

Response: *As suggested, "found" was replaced by "manifested".*

6. P1, 19: What does "closes to -1 oxidation state" mean?

Response: *The sentence, "----- to be attacked by the nucleophilic gold atom which closes to -1 oxidation state." was simply replaced by "----- to be attacked by the nucleophilic gold(-I) atom"*

7. P1, 20: The sentence starting with "We show the bond formation..." does not make any sense. Methylgermane **5** only contains a methyl group at germanium.

Response: *The sentence and the subsequent one were modified as "The reaction of digoldgermane **3a** with electrophilic MeOTf gave the corresponding methylated germane **5**. Digoldgermane **3a** also reacted smoothly with nucleophiles such as Ph₄PCl and CH₃COCl to give the corresponding chloride-addition product **7** and chloro(gold)germane **9**, respectively.*

8. P2, 26: The interest in catalysis, microelectronics, materials and pharmacology is anything but new. What is the point of this claim anyway?

Response: *We wanted to emphasize the significance of gold-metal and metalloid bond complexes but the statements are too general and less important. They were removed in the*

revised manuscript.

9. P2, 29: The “chi” Symbol for electronegativity is displayed as a box in the pdf-file.

Response: *The mistake was corrected.*

10.P2, 35: Delete “firstly”

Response: *It was deleted.*

11.P2, 38: Sentence is nonsense: the lower electronegativity of boron may result in Au-B bonds with boron auride character. It is, however, no prerequisite for Au-B formation at all!

Response: *We agree with the argument of reviewer #1. The paragraph starting with the chemistry of gold compounds ----” was modified largely in the revised manuscript.*

12.P3, 46: “intermediacy of V during these reactions was suggested.”

Response: *Because compound V in the previous manuscript was a proposed reactive intermediate and not characterized, we have decided that the compounds may not be appropriate to be listed in Fig. 1. Another goldgermane was listed in the revised Fig.1 as compound V. In this relation, the paragraph starting with “In a previous paper, -----” were largely modified.*

13. P3, 47: Why “particularly interesting”? All p-block metals and semi metals are less electronegative than gold!

Response: *Electronegativity of Ge is even larger than Si and Sn in group-14 elements. We expected the electronegativity difference may affect the chemistry of bonds of Au-group-14 element. The sentence was modified in the revised manuscript.*

14. P3, 52: “..., isolable digoldgermanes are still unknown to date.” While this statement is probably correct, it would be appropriate to discuss known species with two gold atoms bridged by a single atom. For instance, mu-imido complexes without obvious aurophilic interactions

have been reported by Sharp et al.

Response: *Papers studying single-atom bridged polygold complexes, including the μ_3 -imido trinuclear gold(I) reported by Sharp et al. were stated briefly in the revised manuscript.*

15.P3, 64: “as white solids”

Response: *Corrected.*

16.P3, 65: Delete superfluous “until now”.

Response: *The phrase was deleted.*

17.P4, 69: “can be stored at ambient temperature for a few months [without decomposition].”

Response: *“without decomposition” was added as suggested.*

18.P4, 72: Replace “like each other” by “very similar”

Response: *corrected as suggested.*

19.P4, 75: How does a SLIGHT [bare significant] shortening suggest SIGNIFICANT differences in bonding?

Response: *The word “slightly” is replaced by “somewhat”.*

20.P4, 79: A bond angle of close to 180° does not indicate almost-linearity, it is a manifestation of almost-linearity.

21.P4, 80: Wh invoke VSEPR? Are any lone-pairs involved?

Response to (20) and (21): *the phrase “indicating ---- with the VSEPR theory” was replaced by “manifesting the $^1\text{Ge}-^1\text{Au}-^2\text{Ge}$ and $^1\text{Ge}-^2\text{Au}-\text{P}$ are almost linear, being in accord with the theoretical calculations (vide infra).”*

22.P4, 80: Why INTRINSICALLY perpendicular? They are close to perpendicular, that’s all. This

is presumably due to steric reasons as **3** is not an allene after all.

Response: *The word “intrinsically” is removed and modified the sentence in the revised manuscript.*

23. P4, 84: See comments before. A sum of angles of close to 360° is a manifestation of a planar coordination environment, not evidence.

Response: *The description was modified according to the suggestion.*

24. P4, 85: Only one unique bonding feature?

Response: *“feature” is replaced by “features”.*

25. P5, 96: What is DNMR? Do the authors mean VT-NMR?

Response: *Yes, DNMR is often used as an acronym for Dynamic NMR. Here, however, “VT-NMR” would be more appropriate. “DNMR” was replaced by “VT-NMR”.*

26. P6, 112: Replace evaluated by “estimated”. The equation is a relatively rough approximation.

Response: *The word “evaluated” is replaced by “estimated” as suggested.*

27. P6, 117: The sentence is very similar to the previous sentence. Please reword to avoid repetition.

28. P7, 125: What does the sentence starting with “Au center” mean?

Response to (27) and (28): *The sentences discussing the formation mechanism of **3** were modified in the revised manuscript.*

Reviewer #2 (Remarks to the Author):

The manuscript titled: Unique Digoldgermanes: Structural Characteristics, Dynamic Behaviour and Distinctive Reactions by Wang, Zhen, Li, Kira, Yan, Chang, Huang, and Li can be considered a nice extension of the recent paper by the same authors in Angew. Chem. on digoldstannanes.

In the paper, the authors describe the preparation of an interesting class of compounds where Kira's germylene is coordinated to two gold atoms. One of these gold atoms bears a phosphine ligand, whereas the second one is coordinated by another germylene. The presence of two germylenes and two gold atoms in the compound allows for an interesting dynamic behavior, where the two germylenes switch roles. This "pendulum motion" is nicely studied by VT NMR spectroscopy and also by theoretical calculations. Furthermore, the reactivity of the compound toward nucleophiles and electrophiles was investigated to some extent. Finally, the compound was found to exhibit remarkable catalytic properties for the trimerization of arylcyanates.

1. In the experimental part, the reaction of germylene **1** with Ph₄PCl is described. I did not find an explanation for this in the text and in addition, I have no idea what the source of the proton in the final product is!

***Response:** Thank you for pointing out this question. Actually, the proton may come from the unavoidable trace moisture in the reaction system. Because the product could also be formed by the insertion of HCl to the germylene and because the results are not important in this paper, the description about the reaction was omitted from the manuscript and SI.*

2. I found a number of things that require attention as listed below. Nothing too serious! In line 74 replace "a divalent" by "the divalent".

***Response:** Corrected.*

3. In the sentence starting in line 74 it is stated that "the distance between the divalent germanium (2Ge) and 1Au [2.4146(4) and 2.4089(8) Å for 3a and 3b, respectively] is slightly shorter than those of 1Au-1Ge and 2Au-1Ge bonds, suggesting significant difference in the bonding nature between Ge(IV)-Au and Ge(II)-Au bonds." If there is only a slight effect, how can this suggest a significant

difference??

Response: *The comment is similar to that of Reviewer #1 (2 (n)). The word “slightly” was replaced by “somewhat”.*

4. In line 95 the spectra are a little confused. I suppose what the authors rather mean is confusing.

Response: *“confused” is replaced by “confusing”.*

5. In scheme 2 I wonder if it might be more appropriate to draw the curved arrow of 3 and 3' between the two gold atoms because this interaction leads to 3T. In addition in 3T curved arrows might be drawn to the two Ge atoms.

Response: *The scheme was modified. We thank reviewer #2 for this suggestion.*

6. The next sentence speculates about back bonding between Ge and Au. However, the calculations later disprove this. So why not omit this sentence?

Response: *The related sentence was deleted.*

7. On page 8 the reference to the figure containing effective charges should be to Figure 4 and not 3 (line 158).

Response: *This error was corrected.*

8. The authors own ref 42 Ref. is more likely “Kira, M. et al.” than “Mitsuo, K. et al.”

Response: *In the reference 42, “Mitsuo, K. et al.” has been replaced by “Kira, M. et al.”.*

9. On page 9 it is stated that “no reaction occurs when excess stannylene is added in the solution of 3a.” What “stannylenes” do the authors refer to here (line 172)?

Response: *The mistake is corrected; “stannylene” should be read as “dialkylgermylene I”.*

10. In line 173 replace “N-Heterocyclic” by “N-heterocyclic”

Response: *Corrected.*

11. In line 175 make compound number 3 bold and replace “tow” by “two”.

Response: *Corrected.*

12. In line 176 put a period after “reactions”.

Response: *Corrected.*

13. In line 182 replace Fig. 4 by Fig. 5. Is “Three Ge1-Au1 distance” supposed to mean “The Ge1-Au1 distance”??

Response: *Corrected.*

14. In the caption of Fig. 5 replace “structures” by “structure”.

Response: *Corrected.*

15. In Fig. 7 incorporate compound number 8 for acetyl chloride.

Response: *Acetyl chloride was labeled with “8”.*

16. In line 262 “replace “¹³C” by “13 signals”.

Response: *“signals” was added after ¹³C as suggested.*

REVIEWER COMMENTS

Reviewer #1 (Remarks to the Author):

Li et al. have addressed all issues raised by the reviewers. The paper is now a much more fluent read and linguistic mistakes are considerably less frequent. The manuscript is now recommended for publication in *Nature Communications*. A (now comprehensive) list of further changes of mostly editorial nature follows:

21: "Digoldgermane 3a does not only react..."; "..., but also reacts with..." "...chloride products 7 and 9,..."

24: "...efficient catalyst for the cyclic..."

47: "...have been reported to date,..."

49: "...are known and widely utilized in homogenous catalysis..."

63: "Compounds 3a,b are isolated..."

71: The contradiction is not eliminated by replacing "slightly" by "somewhat"! This insignificant difference does not suggest anything other than the contrary, that is a similar bonding nature. Of course, differences may become apparent in other data...

74: "..., and thus close from linearity, in accord..."

75: "...are almost perpendicular with dihedral angles between..."

78: "..., suggesting any aurophilic interactions to be weak at best. Significant..."

85: Consider replacing "a little confusing" by "complex"

118: Delete article before "DFT"

161: Which N-heterocyclic carbene? Surely, very small NHCs will react?

Reviewer #2 (Remarks to the Author):

I have read the revised version of the digold germane manuscript.

Most of the criticized things have been changed in a satisfactory way. However, I came across a couple of other things, some of the new, some I might have missed before.

- In the abstract replace "product 7 and 9" by "products 7 and 9 by".
- First sentence Full Text: add " $\chi =$ " before "2.54".
- On page 2 the sentence "Gold related compounds bonded to main group metal and metalloid elements have long attracted much attention since then" might be deleted as the following sentence is very similar.
- At the end of page 2 it is mentioned that "...two gold atoms heterolytically coordinated by dialkylgermylene 1". Although I understand what the authors mean I think that the expression "heterolytically coordinated" is a bit awkward.
- At page 4 and on two other occasions I suggest to change "in the NMR time scale" by "on the NMR time scale".
- On page 6 change "The 1Ge-1Au [and 1Ge-2Au] bonds are both covalent formed by the overlap between a 4sp^{3.13} hybrid orbital of 1Ge and a 6s orbitals of 1Au and 2Au; their

Wiberg bond index is 0.6251 and 0.7122 with the occupancy of 1.8740 and 1.8909, respectively.” to “The 1Ge-1Au [and 1Ge-2Au] bonds are both covalent, formed by the overlap between a 4sp^{3.13} hybrid orbital of 1Ge and 6s orbitals of 1Au and 2Au; their Wiberg bond indices are 0.6251 and 0.7122 with the occupancy of 1.8740 and 1.8909, respectively.”

- On page 8 for some strange reason the energy of Int1 is described as “lower” than Int2. This was correct in the original version but needs to be corrected to “higher” now.
- On page 9 the sentence : “Facile isomerization between 3 and 3’ as shown in Scheme 3...” needs to be changed to Facile isomerization between 3 and 3’ as shown in Scheme 2 ...”
- Also on page 9 the sentence “The reaction of 3a with acetyl chloride 8 at ambient temperature ...” does not refer to Fig. 7a but Fig 8a.
- The next sentence should have an “an” before the “acetyl counterion”
- The experimental description of compound 9 gives 5 SiMe₃ signals. It would be nice to have an expansion plot of Fig S42 because to me this looks like three signals.
- In the General procedure for cyclic trimerization reaction the concentration of 3a should be “7.66x10⁻⁴ M” with superscript “-4”.

As I have made clear my personal preference for publication in my initial assessment, I want to express that after correction of the mentioned points raised above I support publication of this nice paper.

We deeply appreciate the two reviewers for their awesome careful and patience comments. They are very valuable for us to revise the manuscript. The manuscript has been revised in light of the reviewers' comments, where the words and sentences highlighted with yellow color are our responses to the comments of reviewer #1 and reviewer #2, respectively.

REVIEWER COMMENTS

Reviewer #1 (Remarks to the Author):

Li et al. have addressed all issues raised by the reviewers. The paper is now a much more fluent read and linguistic mistakes are considerably less frequent. The manuscript is now recommended for publication in Nature Communications. A (now comprehensive) list of further changes of mostly editorial nature follows:

- 1) 21: “Digoldgermane 3a does not only react...”; “..., but also reacts with...” “...chloride products 7 and 9,...”

Response: *The sentence was corrected according to the comment.*

- 2) 24: “...efficient catalyst for the cyclic....”

Response: *The sentence was modified in the revised manuscript.*

- 3) 47: “...have been reported to date,...”

Response: *The word “known” was replaced by “reported” in the light of reviewer’s suggestion.*

- 4) 49: “...are known and widely utilized in homogenous catalysis...”

Response: *The sentence was modified in the light of reviewer’s comment.*

- 5) 63: “Compounds 3a,b are isolated...”

Response: *The sentence was modified according to the suggestion.*

- 6) 71: The contradiction is not eliminated by replacing “slightly” by “somewhat”! This insignificant difference does not suggest anything other than the contrary, that is a similar bonding nature. Of course, differences may become apparent in other data...

Response: *We agree with the argument of reviewer #1. The sentence is modified in the revise*

manuscript, where “significant difference bonding nature.....” was replaced by “a similar bonding nature....”

7) 74: “..., and thus close from linearity, in accord...”

Response: *The sentence was modified according to the suggestion.*

8) 75: “...are almost perpendicular with dihedral angles between...”

Response: *The sentence was modified according to the suggestion.*

9) 78: “..., suggesting any aurophilic interactions to be weak at best. Significant...”

Response: *The sentence was modified according to the suggestion in the revise manuscript.*

10) 85: Consider replacing “a little confusing” by “complex”

Response: *The word “a little confusing” was replaced by “complex” in the light of reviewer’s comment.*

11) 118: Delete article before “DFT”

Response: *The word “the” was removed as suggested.*

12) 161: Which N-heterocyclic carbene? Surely, very small NHCs will react?

Response: *The word “N-heterocyclic carbene” was explained in detail, after which “1,3-bis(2,6-diisopropylphenyl)imidazol-2-ylidene” was added as comment.*

Reviewer #2 (Remarks to the Author):

I have read the revised version of the digold germane manuscript.

Most of the criticized things have been changed in a satisfactory way. However, I came across a couple of other thing, some of the new, some I might have missed before.

- 1) In the abstract replace “product 7 and 9” by “products 7 and 9 by”.

Response: *The comment is similar to that of Reviewer #1 (1). The word “slightly” was replaced by “somewhat”. The sentence was corrected according to the comment.*

- 2) First sentence Full Text: add “ $\chi =$ ” before “2.54”.

Response: *The symbol for electronegativity was added.*

- 3) On page 2 the sentence “Gold related compounds bonded to main group metal and metalloid elements have long attracted much attention since then” might be deleted as the following sentence is very similar.

Response: *The sentence was removed according to the comment.*

- 4) At the end of page 2 its is mentioned that “...two gold atoms heterolytically coordinated by dialkylgermylene 1”. Although I understand what the authors mean I think that the expression “heterolytically coordinated “ is a bit awkward.

Response: *The word “heterolytically” was removed in the revised manuscript.*

- 5) At page 4 and on two other occasions I suggest to change “in the NMR time scale” by “on the NMR time scale”.

Response: *The preposition “in” was replaced by “on” at page 1, 4, and 10, respectively.*

- 6) On page 6 change “The 1Ge-1Au [and 1Ge-2Au] bonds are both covalent formed by the overlap between a 4sp^{3.13} hybrid orbital of 1Ge and a 6s orbitals of 1Au and 2Au; their Wiberg bond index is 0.6251 and 0.7122 with the occupancy of 1.8740 and 1.8909, respectively.” to “The 1Ge-1Au [and 1Ge-2Au] bonds are both covalent, formed by the overlap between a 4sp^{3.13} hybrid orbital of 1Ge and 6s orbitals of 1Au and 2Au; their Wiberg bond indices are 0.6251 and 0.7122 with the

occupancy of 1.8740 and 1.8909, respectively.”

Response: *The sentence was modified according to the suggestion.*

- 7) On page 8 for some strange reason the energy of Int1 is described as “lower” than Int2. This was correct in the original version but needs to be corrected to “higher” now.

Response: *Corrected.*

- 8) On page 9 the sentence: “Facile isomerization between 3 and 3’ as shown in Scheme 3...” needs to be changed to Facile isomerization between 3 and 3’ as shown in Scheme 2 ...”

Response: *Corrected.*

- 9) Also on page 9 the sentence “The reaction of 3a with acetyl chloride 8 at ambient temperature ...” does not refer to Fig. 7a but Fig 8a.

Response: *Corrected.*

- 10) The next sentence should have an “an” before the “acetyl counterion”

Response: *The article “an” was added.*

- 11) The experimental description of compound 9 gives 5 SiMe₃ signals. It would be nice to have an expansion plot of Fig S42 because to me this looks like three signals.

Response: *The Fig S42 was modified according to the suggestion of reviewer 2#.*

- 12) In the General procedure for cyclic trimerization reaction the concentration of 3a should be “7.66x10⁻⁴ M” with superscript “-4”.

Response: *Corrected.*